# Study of Abrasive Water Jet Machining as a Texturing Operation for Thin Aluminium Alloy UNS A92024

**DOI:** 10.3390/ma16103843

**Published:** 2023-05-19

**Authors:** Fermin Bañon, Alejandro Sambruno, Pedro F. Mayuet, Álvaro Gómez-Parra

**Affiliations:** 1Department of Civil, Materials and Manufacturing Engineering, University of Malaga, C/Dr. Ortiz Ramos s/n, 9071 Malaga, Spain; 2School of Engineering, Mechanical Engineering and Industrial Design Department, University of Cadiz, Av. Universidad de Cadiz 10, Puerto Real, 11519 Cadiz, Spain; alejandro.sambruno@uca.es (A.S.); pedro.mayuet@uca.es (P.F.M.); alvaro.gomez@uca.es (Á.G.-P.)

**Keywords:** AWJM, texturing, wettability, waterjet, surface quality, abrasive

## Abstract

Surface modification of metallic alloys can create hydrophilic or hydrophobic surfaces that enhance the functional performance of the material. For example, hydrophilic surfaces have improved wettability, which improves mechanical anchorage in adhesive bonding operations. This wettability is directly related to the type of texture created on the surface and the roughness obtained after the surface modification process. This paper presents the use of abrasive water jetting as an optimal technology for the surface modification of metal alloys. A correct combination of high traverse speeds at low hydraulic pressures minimises the power of the water jet and allows for the removal of small layers of material. The erosive nature of the material removal mechanism creates a high surface roughness, which increases its surface activation. In this way, the influence of texturing with and without abrasive has been evaluated, reaching combinations where the absence of abrasive particles can produce surfaces of interest. In the results obtained, the influence of the most relevant texturing parameters between hydraulic pressure, traverse speed, abrasive flow and spacing has been determined. This has allowed a relationship to be established between these variables and surface quality in terms of Sa, Sz and Sk, as well as wettability.

## 1. Introduction

Today’s industry and the aerospace sector are characterised by continuous optimisation of the production process. In this sense, materials and processes that have been used in the industry for years are still being studied to find improvements [1,2]. This is the case with UNS A92024-T3 aluminium, a material of great importance in the construction of primary and secondary aircraft components due to characteristics such as its formability. One of the main reasons why this type of material continues to be studied is the trade-off between the weight and mechanical properties required in the aerospace sector. Therefore, the importance of the treatment of strategic thin materials is a fundamental factor for various industrial sectors, as is the case with aluminium alloys [3,4,5].

On the other hand, it is well known that mechanical joints are the most widely used in the aerospace sector, but adhesive joints are also replacing the main mechanical joints for several reasons. The use of adhesive bonds between materials of the same or different natures is a key parameter for reducing the weight of the component [6]. In addition, the presence of a layer that provides continuity between the materials that make up the stack can offer advantages, such as the elimination of interlayer defects or wide adaptability to materials, as well as minimising the time required for quality control or avoiding adhesion due to contaminants [7,8,9]. For this reason, this type of joint is constantly being studied and is of great interest to several industrial sectors.

It is very important to analyse the textured surface to ensure the mechanical anchorage created between the bonding materials. In this respect, several publications emphasise the importance of surface activation [10] in obtaining a hydrophilic surface that allows the adhesive to spread over the entire surface [11,12]. The surface finish and the wettability of the surface are the two most influential variables when analysing the results. Regarding the wettability parameter, the angle created between a liquid ball and the textured surface should be analysed to check the tension created between the elements and thus determine whether the surface is hydrophilic or hydrophobic. The generation of very deep peaks along the textured surface can be negative, as this effect could favour the appearance of bubbles at the time of bonding [13,14,15].

On the other hand, it is well known that one of the most frequently used parameters in the analysis of the surface finish is the arithmetic mean roughness or average roughness parameter (Ra). However, since the finish to be analysed in texturing involves a much larger surface area, it makes more sense to use surface quality parameters. Among the most relevant are the average surface roughness parameter (Sa), maximum surface height (Sz) and depth of core roughness (Sk). Thanks to these types of surface parameters, a truer compression of reality can be carried out. A poorer surface finish or higher roughness values can improve the anchorage of the liquid or adhesive.

There are several manufacturing processes capable of texturing the surface of the material. Some of those that have been investigated include laser texturing and shot peening [16,17,18]. The energy required by laser equipment to perform texturing is high, which can increase process costs. In addition, the area allowed by the laser equipment to perform texturing is usually small and the diameter of the spot is in the range of 60 μm. Therefore, the time required to perform laser texturing of a surface is usually high [19]. In addition, the temperature concentration generated to texture the surface can cause thermal defects in the results.

Another technology of interest is shot blasting. This is a widely used process for making surface changes to materials. However, this process is characterised by being applied manually, so it can be difficult to control the distance at which the particles hit the material. This characteristic is a handicap for the process, as it does not guarantee homogeneity along the textured surface, i.e., the volume of particles interacting with the surface is not controlled [10]. This process can create excessive roughness, resulting in voids or internal bubbles on the surface.

Abrasive waterjet equipment has been the subject of many studies in recent years [20,21,22]. Its ability to work elements with geometries and/or materials that are difficult to machine has made it one of the most widely used processes in the aerospace sector. It is a versatile tool where, if the process is properly controlled, it is possible to machine from very soft to very hard materials [23]. All these characteristics mean that abrasive waterjet machining offers possibilities that have not yet been fully exploited, such as its usefulness for the surface modification of materials [24].

Among other advantages, Abrasive Water Jet Machining (AWJM) is a faster process than other texturing technologies. It is also capable of texturing a large area simultaneously. It is a kinetic energy-based process where a high combination of high traverse speeds at low pressures, with or without abrasive, and long blast-to-work distances minimises the penetration effect from cutting to gouging or surface modification.

These characteristics provide greater robustness and homogeneity of the textured surface compared to processes such as laser and shot blasting. Another feature achieved by using abrasive waterjet texturing technology is that the distance between the focusing tube and the material is guaranteed to be the same over the entire surface [25]. In addition, the spindle speed can be high and constant. This feature, together with the numerical control of the equipment, makes it possible to modify the strategy followed by the cutting head and to analyse the influence on the results by generating different patterns. The spot diameter is larger than in the laser texturing process. This characteristic, together with the overlapping of trajectories, makes it possible to produce more controlled patterns than with shot peening and over larger areas than with laser texturing.

In this sense, the most influential parameters in abrasive waterjet texturing are [26,27]: abrasive mass flow rate (AMFR), path overlap, focusing tube-to-material distance (SOD) and traverse speed (TS). With AWJT technology, values similar to those obtained with laser texturing can be achieved [28].

Pahuja et al. [29] used abrasive waterjet equipment to create surface modifications on 3D printed metal parts. The use of high-pressure water allowed them to control the depth of erosion, improving surface roughness and subsurface microhardness. Sourd et al. [30] also used AWJT technology as a surface preparation technique for bonding 3D printed carbon fibre reinforced composites (CFRP). The tests performed were analysed using the surface quality parameters Ra and Sa, as well as the crater volume created after texturing.

Other studies, such as Ibrahim et al. [31], in their research, showed how AWJM machining can be used to produce aluminium oxide micromoulds, achieving freestanding structures up to 435 μm in height, while still maintaining an acceptable surface finish (1.51 μm Ra). In addition, Popan et al. [32] proposed some strategies for waterjet manufacturing, where they agreed that the right machining strategies and proper process parameters in waterjet milling is a good solution for producing flat surfaces, profiles and slots. Hejjaji et al. [33], in their research, investigated the influence of controlled depth abrasive waterjet milling on the fatigue of carbon/epoxy composites. Hejjaji et al. showed how machined specimens with high crater volume exhibited inferior fatigue behaviour, and X-ray tomography revealed that crack/fracture initiation occurred from the crater edges.

The number of studies on the abrasive waterjet texturing of metallic alloys is limited. However, it is intended that this technology will control the results obtained in materials of low thickness without causing defects in the final textured parts. In this way, it is possible to control the effective material reduction after the process, i.e., wall thinning in strategic materials [34].

In summary, this paper describes the abrasive waterjet texturing of a thin sheet of UNS A92024-T3 aluminium alloy. By combining abrasive mass flow rate (AMFR), overlap, focusing tube-to-material distance (SOD) and head feed rate (TS), the textured surface is analysed in terms of surface finish and wettability. In this way, the influence of the parameters will be established, as well as predictive models of interest to the industry and control of the final thickness obtained.

## 2. Materials and Methods

To achieve the objectives set out above, an experimental methodology was proposed.

### 2.1. Material and Machining Process

The aluminium alloy was UNS A92024-T3, as shown in Table 1 and Table 2. To corroborate the composition of the alloy, X-ray photoelectron spectroscopy tests (XPS) were carried out. In this case, Spectrolab M12 equipment (Ametek, Kleve, Germany) was used.

In terms of material dimensions, a 500 × 500 × 2 mm^3^ plate was used for this experiment. From this raw material, 24 specimens were cut. In each, specimens of dimensions 50 × 20 × 2 mm^3^ were obtained. These dimensions refer to the pre-machined aluminium test piece. A clamping area of 10 × 20 mm^2^ was left at each end, resulting in a textured area of 30 × 20 mm^2^.

The parameters to carry out the surface texturing of these specimens are shown in Table 3, based on previous studies carried out [28]. For these purposes, a TCI Cutting (TCI Cutting, Valencia, Spain) waterjet machine was used. The texturing strategy used was a back-and-forth process. The CAD/CAM software used to programme the trajectories was LANTEK.

Moreover, during the machining process, the parameters shown in Table 4 were kept constant. A texturing strategy of parallel lines with a constant distance between them was established. This distance depended on the overlap parameter in reference to the focusing tube diameter (25%: 0.19 mm; 50%: 0.36 mm). A separation distance between the start of the water jet path and the start of the material of 20 mm was also set. This allowed a constant flow of water and abrasive particles to be obtained and the surface to be textured at a constant travel speed.

A 120 mesh Indian garnet abrasive size with a rounding factor of 0.8 was used during texturing operation and its chemical composition, as shown in Figure 1.

Finally, after texturing, the samples were removed from the abrasive waterjet machine, cleaned and dried with compressed air.

### 2.2. Test Evaluation

Surface integrity was evaluated in terms of geometric properties. In terms of microgeometric properties, the average surface roughness (Sa, Sz, Sk) was measured. In abrasive waterjet cutting, the loss of kinetic energy in the waterjet created two or three regions of different surface quality. This means that an evaluation by Ra or Sa did not reflect reality. In texturing operations, however, the surface type was more homogeneous, as a very constant average depth value was obtained. This allowed the Sa parameter to be more representative. However, in order not to lose any information, it was decided to complement the study of surface quality with the Sz and Sk parameters in order to obtain a better understanding of the type of surface obtained.

An Alicona InfiniteFocus G5 (Alicona Imaging GmbH, Graz, Austria) variable focus microscope was used for this purpose. This equipment allowed high-precision 3D optical measurements and the generation of 3D models in STL format for further processing. For the evaluation of surface quality, the ISO 25178 [35] standard was followed, applying a Gaussian filter for flat surfaces under ISO 16610-61 [36]. To avoid noise due to the start of the texturing operation, an area that did not include the edges of the material was evaluated (Figure 2).

The 3D models in STL format were used to achieve maximum depth after texturing. PolyWorks software was used to achieve this.

In this sense, 35 control points were placed in each textured area, resulting from the intersection of 5 planes in the direction in which the sample measured 20 mm, and 7 planes in the direction in which the sample measured 50 mm.

Each intersection was projected onto the textured surface. The depth measurement result was obtained by measuring the point projected onto the surface with a plane containing the untextured aluminium surface of each sample (Figure 3).

Wettability tests were carried out to evaluate the contact angle on textured surfaces [37,38]. A drop of distilled water was deposited on each surface and evaluated using a phase contact angle measurement system consisting of a high-resolution CCD camera positioned on the axis traversing the drop, while a back-illumination point provided contrast to capture the geometry (Figure 4).

Image processing software was then used to measure the tangent contact angle in the geometry of the deposited droplet to assess whether it exhibited hydrophobic or hydrophilic behaviour. Three contact angle measurements were obtained to generate a mean value with deviations close to 0.1.

### 2.3. Data Processing

The analysis of the results was carried out in three main steps. The first was to identify the trends between the cutting parameters and the variables studied. The second step was based on the quantification of the weight of each cutting parameter. To this end, an analysis of variance (ANOVA) with a 95% confidence interval was carried out.

Finally, the results of the ANOVA study were considered in the graphical representation of the contour plots. The statistical analysis was carried out using MiniTab statistical software v18.

## 3. Results

### 3.1. Experimental Results in Abrasive Waterjet Texturing

The evaluation of the surface quality in terms of Sa, Sz and Sk was carried out according to the steps indicated in the methodology section. First, the results obtained for the Sa parameter of the different combinations of texturing parameters are shown in Figure 5.

In the case of a traverse speed of 4000 mm/min, without the use of abrasives, a similar behaviour was observed for SoD of 10 and 30 mm with overlaps of 25% and 50%. In fact, increasing the overlap parameter had no effect on the roughness obtained, since it was the SoD parameter that determined the surface quality of the process. The higher the SoD, the better the surface quality expressed in Sa. However, in the case of a SoD of 50 mm, there was a variation in which a higher roughness was obtained for an overlap of 50% compared to that obtained for an overlap of 25%. In these cases, the combination of increased distance to the part and overlap seemed to influence the increase in roughness and, therefore, the deterioration of the surface quality, Sa. This can be explained by the dispersion of the water jet on the surface, which, in the case of the 50% overlap parameter, did not have sufficient energy to produce a homogeneous surface in terms of roughness.

As can be seen for the TS 6000 without abrasives in the 25% and 50% overlap cases, there were two well-differentiated results, with no influence of the distance from the workpiece on the roughness result (Figure 6a). In fact, for the 25% overlap, an average Sa of about 10 microns was obtained, while for the 50% overlap, the Sa was improved, generally obtaining a value of about 2 µm for all SoD. We can therefore say that overlap plays a very decisive role in terms of Sa.

For a TS of 6000 mm/min, increasing the percentage overlap generated a reduction in Sa values as the SoD parameter increased.

In this case, due to the increase in the head traverse speed, the time that the beam was on the surface was reduced; for an overlap of 25% (see Figure 6b), the energy of the beam on the surface was reduced, which, together with the low percentage of overlap, resulted in surfaces with greater roughness. However, for an overlap of 50%, the results obtained for the three SoDs studied were the best in terms of Sa. In this case, the energy delivered by the water jet with the 50% overlap parameter resulted in more homogeneous surfaces and a higher surface quality, which reduced the machining time due to the increase in speed.

In the abrasive texturing condition, the effect of the SoD parameter was visible in the results obtained with a reduction in the Sa parameter and a 25% overlap. For this condition, increasing the SoD from 10 mm to 50 mm reduced the Sa values to results similar to those generally obtained with 50% overlap, regardless of the SoD value set.

It can be concluded that reduced overlap increased the effect of the stand-off distance. For this condition, increasing this parameter minimised the kinetic energy of the water jet by reducing the overexposure of the surface to the abrasive particles. This allowed values close to 20 µm to be obtained.

In turn, these values were obtained with a variation of 5 µm for a 50% overlap, regardless of the SoD parameter set. Therefore, for an abrasive flow rate of 110 g/min, an improvement in surface quality was obtained under these conditions, and by setting the overlap to 50%, an increase in process performance was obtained by reducing the operating time.

The effect of the abrasive was also highly significant in terms of the amount of material removed and the depth of penetration of the water jet. This was a key factor since, for correct texturing and subsequent application of the adhesive, the final thickness of the material must be constant. The average depth and its deviation for the tests carried out are shown in Figure 7.

Combinations of traverse speeds close to 6000 mm/min and abrasive flows interacted longer with the material, increasing the amount of material removed and the depth of penetration. This can also lead to more turbulence and instability of the process, which can be seen in the increased deviation of the results.

In contrast, texturing with pure water minimised the amount of material removed and showed minimal deviation. This would indicate greater robustness of the process and greater control over the amount of material to be removed. Bearing in mind the reduced thickness of the material under study and the nature of the machining process, it should be noted that water jet texturing without abrasive generated controlled thinning of the walls.

The results corresponding to parameter Sz are shown in Figure 8.

An increase in the distance between each pass increased the maximum distance between the peak and valley on the evaluated surface. By increasing this distance, the overlap of the area affected by the water jet was reduced. Due to the divergence of the jet, the penetration capacity was of the parabolic type, generating greater depth in the central part of the cavity. By separating these cavities, more defined grooves were formed, increasing this value, as observed in Figure 9. This is of great interest for subsequent bonding applications to guarantee higher mechanical anchorage.

Increasing the traverse speed reduces the effect of the distance between passes in Sz. This was because the time the surface was exposed to the water jet was significantly reduced. This reduction in exposure time minimised the depth of surface modification, resulting in a smoother and more homogeneous surface.

As for the SoD parameter, it did not seem to have a significant influence on the process or the formation of the surface roughness Sz. This was observed both for the increase of TS and for the tests carried out with and without abrasive, although the Sz values seemed to be more stabilised at higher speeds. This was in good agreement with those discussed in the previous paragraph. Furthermore, it is important to highlight the difference in magnitude of the Sz value depending on the abrasive, taking higher values when it was used, as is the case with Sa. Thus, by minimising the dispersion of the water jet and increasing the erosive capacity, a rougher surface was produced.

The results corresponding to the Sk parameter are shown in Figure 10.

Sk is a parameter within the category of functional performance parameters in the evaluation of the surface quality of areas [39]. This is related to the functionality of the surface for its performance. Thus, the higher the Sk values, the higher the resistance to mechanical stress and wear. Sk represents the core roughness of the surface over which a load may be distributed after the surface has been run in [40].

A reduction in Sk was seen with increasing interpass distance and with increasing part-blast distance. This may be due to the interaction between the water particle flow and the surface. Increasing these parameters reduced kinetic energy and their interaction. This softened the surface and may have worsened its mechanical behaviour.

If a surface with good tribological behaviour is required, the application of abrasive is essential. This parameter increased the value of Sk from 20 microns to about 140.

Regarding traverse speed, its influence was remarkable when there was no abrasive mass flow. By eliminating this, the surface modification capability of the water jet was drastically reduced. By increasing the displacement velocity in these conditions, the interaction time between the water particles and the surface was minimized, which was observed in the reduction of Sk values with values close to 5 μm.

On the other hand, when abrasive particles were included, the highest Sk values were obtained for an overlap of 25%. For both displacement velocities, a reduction in this parameter was observed. Thus, by reducing the separation between passes, the interaction between the water flow and the abrasive particles with the surface increased. This results in overlapping zones affected by the water flow, obtaining a more homogeneous rough surface and increasing the parameter Sk.

### 3.2. Statistical Analysis of Experimental Results and Contour Plots

Table 5 shows the ANOVA statistical analysis used to determine the statistical influence of the process parameters on the variables studied. Thus, depending on the surface quality parameter to be obtained and evaluated, different process parameters were the most relevant for optimising the process.

Regarding the Sa parameter, which is equivalent to the Ra parameter, only the abrasive mass flow rate had a significant influence. Increasing the penetration capacity by mixing the abrasive particles with water particles allowed deeper craters to be created on the surface, as well as greater penetration into the material itself. This significantly increased the values of Sa, as mentioned above.

Regarding the Sz parameter, which is equivalent to the Rz parameter and is of interest for adhesive applications on the surface itself, the displacement speed and the overlap of the water jet were the main parameters. A correct combination of the two makes it possible to increase the maximum depth on the entire surface, which can allow better anchorage of adhesives or paints later.

Finally, the functional parameter Sk, equivalent to the Rk parameter and related to surface integrity and fluid retention in tribological operations, depends mainly on the abrasive flow, followed by the overlap parameter and displacement velocity. It is noteworthy that for the three parameters studied, there was no significant influence of the distance from the workpiece. This may be since the values obtained in this study were very high. This drastically reduced the kinetic energy of the water jet and its opening, so that between distances of 10 and 50 mm, there was no significant variation in the loss of kinetic energy. Smaller values of the SoD parameter could be of interest for future studies, with values close to 3 mm, as in cutting operations.

In combination with the ANOVA analysis, a series of predictive mathematical models based on a response surface (Figure 11) with settings were obtained. From these models, a series of contour plots were obtained relating the studied variables to the main process parameters according to the results obtained in the ANOVA analysis. The predictive models obtained for Sa, Sz and Sk are shown in Equations (1)–(3).
(1)Sa=26−1.72 Step −0.0073 TS+2.20 SoD+1.665 AMFR+0.000430 Step ∗TS −0.0179 Step ∗SoD                    − 0.0139 Step ∗AMFR−0.000306 TS ∗SoD−0.000176 TS ∗AMFR +0.00312 SoD∗AMFR
(2)Sz=−1625+71.4 Step +0.244 TS+7.01 SoD+2.36 AMFR−0.01079 Step ∗TS −0.291 Step ∗SoD                  + 0.0190 Step ∗AMFR+0.00122 TS ∗SoD−0.000361 TS ∗AMFR−0.0492 SoD∗AMFR
(3)Sk=67.6−1.15 Step −0.0070 TS−0.28 SoD+2.141 AMFR+0.000146 Step ∗TS −0.0018 Step ∗SoD                    − 0.01075 Step ∗AMFR+0.000065 TS ∗SoD−0.000196 TS∗AMFR −0.00271 SoD∗AMFR

The relationship between the main input parameters and the three surface quality evaluation parameters studied as a function of the results of the ANOVA analysis was shown. For the 3 cases, the separation between passes was a determining factor that considerably modifies the final surface quality. Thus, values of traverse speed close to 4000 mm/min and when texturing with abrasive particles was used, the influence of the overlap parameter increased, modifying the surface quality.

The fit of the models was 78.91% for the Sa parameter, 94.20% for the Sz parameter and 96.54% for the Sk parameter.

### 3.3. Wettability

The angle formed by the droplet with the metal surface gives an approximation of the level of adhesiveness of the material. The texturing process seeks to increase the roughness of the surface by varying the value of the contact angle [41]. In this case, the aim was to make the surface as hydrophilic as possible (Figure 12).

In a first observation of the results (Figure 13), it can be seen that the tests carried out without abrasive showed lower contact angle values. This may be due to the fact that the textured surface without abrasive has a lower and more homogeneous roughness result, which may favour a hydrophilic contact surface [42]. On the contrary, in tests with abrasive conditions that were more irregular and rougher, the results showed contact angles close to 120°, revealing the more hydrophobic character of the textured surface. This trend was also directly observed with the overlap selected for each test, where the highest values were recorded for paths with 25% overlap between blast displacements. An increase in overlap resulted in much lower contact angle values as jet exposure decreased.

It should be noted that for the minimum jet-workpiece distance studied (10 mm), a transverse speed of 4000 mm/min and an overlap between passes of 25%, the surface created was hydrophilic with a contact angle of 84.93°.

Figure 14 shows the marginal interaction plots for the parameters TS, SOD, AMFR and overlap. In general, a 50% overlap favoured the formation of smaller contact angles when interacting with the three input variables. Specifically, this formation coincided with tests carried out without abrasion and at high speeds.

It was observed that if the overlap was fixed at 0.19 mm, an increase in the transverse speed led to an increase in the contact angle. Conversely, if the overlap value was doubled, i.e., 0.38 mm, the opposite occurred. Increasing the transverse speed resulted in smaller contact angles.

In the specimens machined with an abrasive water jet, it was verified that, as the value of the traverse speed increased, the value of the contact angle increased. These results coincided with those of other studies [18], in which, although the textured material was s275 steel, as the traverse speed increased, the contact angle also tended to increase.

In the case of the abrasive water jet, the effect of overlap was reduced. The abrasive decreased the surface quality of the piece, i.e., it decreased its roughness and, as can be seen, all surfaces generated were hydrophobic except for the first specimen studied, which had a contact angle of 84.93°. This may be due to the fact that by reducing the speed of displacement, the particles that erode the surface generated greater roughness, which caused the liquid to spread over a wider area.

In order to further investigate the relationship between the selected variables, Figure 14 shows the standardised effects pareto diagram combining the different parameters for a 95% confidence interval. The results showed that the linear factor showing the highest degree of significance was abrasive, as explained in Figure 15. Conversely, the linear factor that had the least influence on the contact angle was SoD. On the other hand, when analysing the interaction of two factors, the only combination that showed significance on the response variable was the overlap with TS. This is due to the relationship already studied in roughness of these two parameters, where it is known that a higher jet velocity can distort the final impact diameter on the passes and the overlap of the passes.

The main effects graph, Figure 16, coincides with those explained in the previous paragraph because it shows that the most determining factor in the variation of the angle is the abrasive, with differences of up to 20° depending on the influence of the rest of the tests. Subsequently, the overlap was the next most influential parameter in the angle variation, with differences per test of up to 10°. Finally, it seemed that TS and SoD were the least influential parameters.

This agreed with the ANOVA analysis (Table 6), which coincided with the experimental results. Thus, the most relevant parameter was abrasive flux. A correct selection of this flux modified the aluminium surface from a hydrophobic to a hydrophilic surface. Moreover, its importance in the final roughness obtained was corroborated.

Finally, the contour plots of the most influential parameters (AMFR and overlap) in the formation of the contact angle are shown. Figure 17 shows that for TS = 4000 mm/min and SoD = 30 mm, contact angle values of 95° were obtained in the absence of abrasive and an overlap of 25%. Similarly, with TS = 6000 mm/min, values of 85 in the absence of abrasive and 50% overlap were obtained. The difference in droplet angle formation was remarkable with abrasive levels AMFR = 0 g/min and AMFR = 110 g/min where the least contact angles were 85° and 102°, respectively, for a SoD = 30 mm.

## 4. Conclusions

A study was carried out on the texturing of thin sheets of UNS A92024-T3 alloy using a waterjet cutting machine with and without abrasives. The main conclusions are as follows:

The addition of abrasive particles was a key factor in the texturing of thin aluminium alloys. Abrasive particles increased the erosion and penetration capability of the flow, resulting in a significant increase in the Sa, Sz and Sk results, as well as the mean depth. However, similar roughness was achieved between texturing with and without abrasive at a traverse speed of 6000 mm/min. This resulted in less interaction with the water flow, allowing similar results to be achieved between the two processes.

In terms of average depth, the robustness of the process in texturing 2 mm thick aluminium alloys stands out. The results obtained showed very small deviations and depths close to 0.025 mm in the case of texturing without abrasive and without causing deformations in the material. This could be a very interesting line for controlled thickness reduction operations as an alternative to other operations.

To improve the bond quality of the future adhesive, more defined voids were observed by increasing the overlap parameter between passes from 25% to 50%. This makes it possible to take advantage of the parabolic geometry resulting from the interaction between the water jet flow and the surface and to define a wavy pattern.

An ANOVA analysis was used to determine the most significant parameters among the variables studied. Thus, the abrasive mass flow rate (AMFR) was the most decisive parameter in the results obtained for Sa, Sk and wettability. On the other hand, the Sz parameter was mainly defined by the displacement speed, which allowed greater or lesser interaction with the surface. This is of interest because for future adhesive applications, high Sz values may allow for higher anchorage.

In terms of wettability, no defined hydrophilic or hydrophobic surface type was observed. This means that there does not seem to be a correlation between surface quality and wettability in the tests carried out. Values between 80° and 100° were obtained. However, it was observed that an increase in the displacement speed increased the contact angle. Increasing this parameter reduced the interaction time between the water and the abrasive particles, which could reduce the surface activation of aluminium.

Finally, based on the experimental results, a series of predictive models for surface quality and wettability were obtained with fits between 80% and 96%, which may be of interest for future tests.

The evaluation of the surface quality after abrasive water jet texturing operations has shown interesting results. This will be evaluated in the future to relate it to the mechanical behaviour of the study material.

## Figures and Tables

**Figure 1 materials-16-03843-f001:**
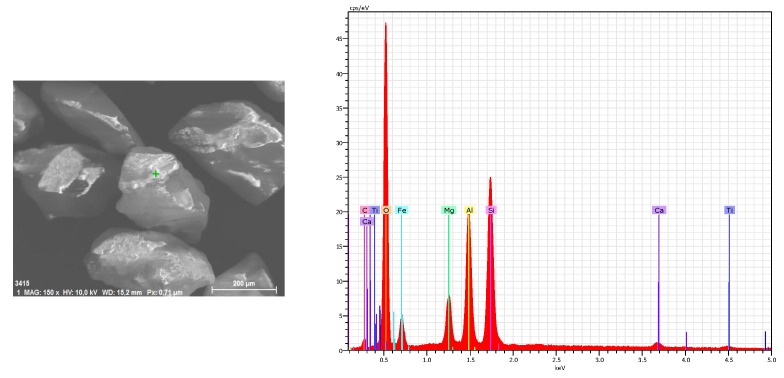
Type of abrasive particles used in the study and chemical composition.

**Figure 2 materials-16-03843-f002:**
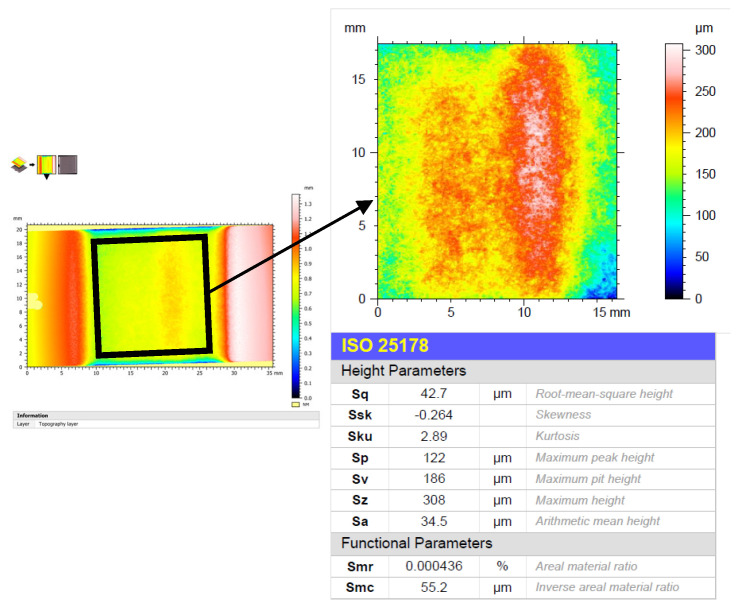
Visual schematic of surface quality assessment methodology using the ISO 25178 standard.

**Figure 3 materials-16-03843-f003:**
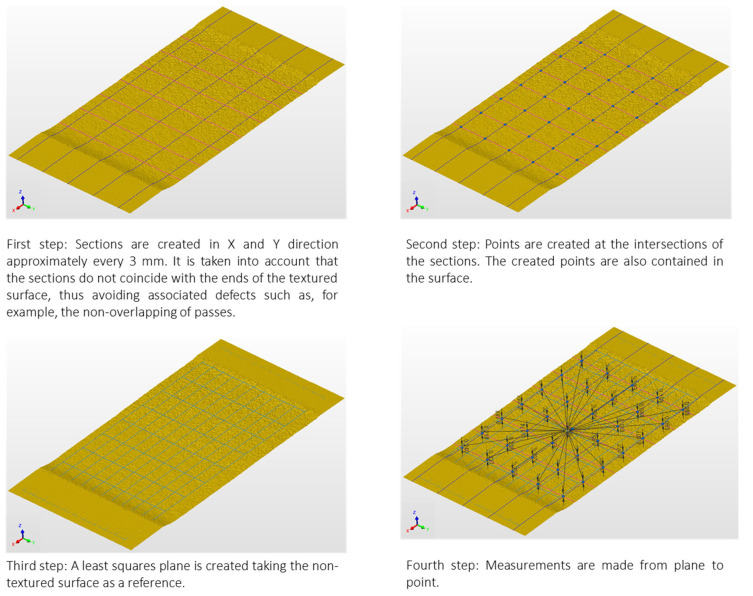
Methodology for evaluation of the penetration depth of the water jet generated after aluminium alloy texturing.

**Figure 4 materials-16-03843-f004:**
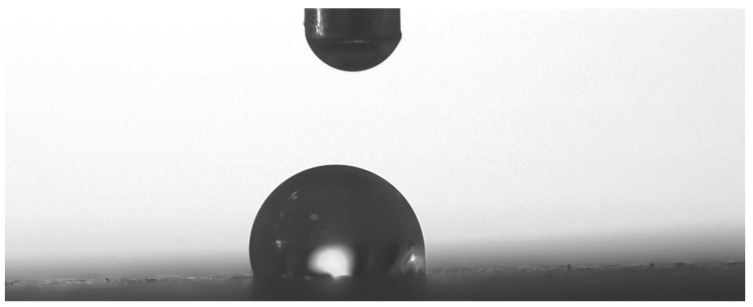
Evaluation of wettability after water jet texturing by assessing the contact angle when depositing a distilled water drop.

**Figure 5 materials-16-03843-f005:**
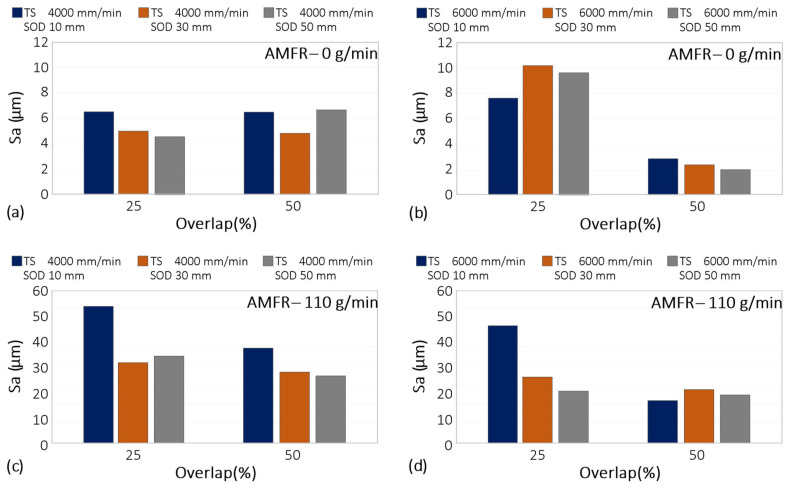
Experimental results for Sa: (**a**) Influence of overlap and SoD for a TS of 4000 mm/min and AMFR 0 g/min; (**b**) Influence of overlap and SoD for a TS of 6000 mm/min and AMFR 0 g/min; (**c**) Influence of overlap and SoD for a TS of 4000 mm/min and AMFR 110 g/min; (**d**) Influence of overlap and SoD for an TS of 6000 mm/min and AMFR 110 g/min.

**Figure 6 materials-16-03843-f006:**
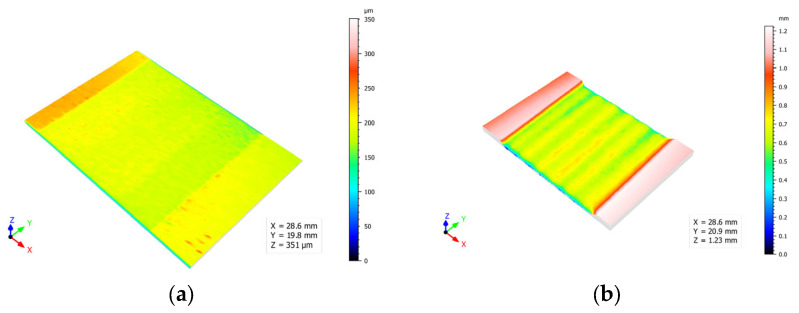
Type of surface obtained as a function of the abrasive application in the texturing process (TS: 4000 mm/min; Overlap: 25%; SoD: 10 mm): (**a**) AMFR: 0 g/min; (**b**) AMFR: 110 g/min.

**Figure 7 materials-16-03843-f007:**
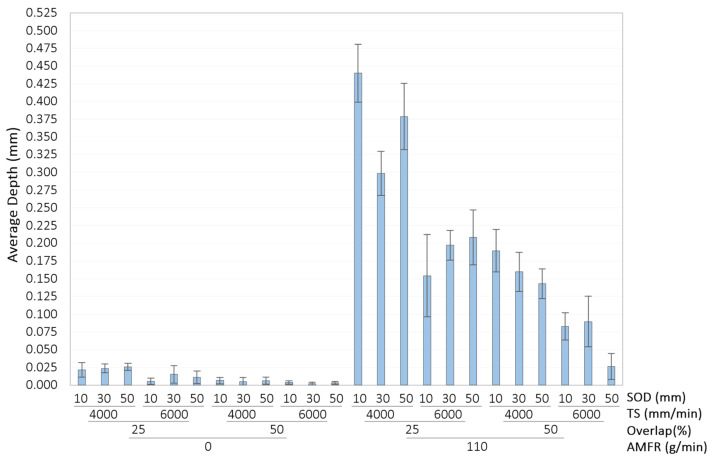
Average depth of water jet penetration and its deviations for the water jet texturing tests performed.

**Figure 8 materials-16-03843-f008:**
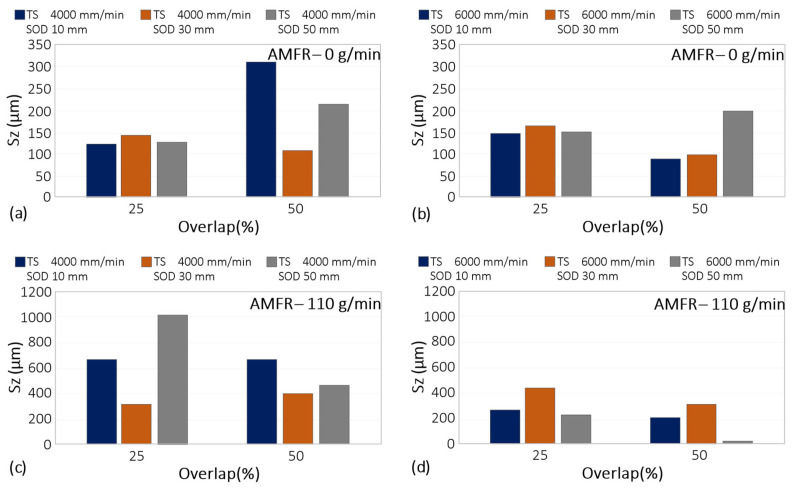
Experimental results for Sz: (**a**) Influence of overlap and SoD for a TS of 4000 mm/min and AMFR 0 g/min; (**b**) Influence of overlap and SoD for a TS of 6000 mm/min and AMFR 0 g/min; (**c**) Influence of overlap and SoD for a TS of 4000 mm/min and AMFR 110 g/min; (**d**) Influence of overlap and SoD for an TS of 6000 mm/min and AMFR 110 g/min.

**Figure 9 materials-16-03843-f009:**
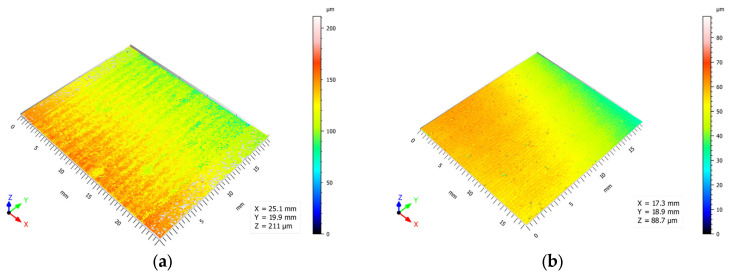
Influence of the overlap parameter on the textured surface (TS: 6000 mm/min; AMFR: 0 g/min; SoD: 10 mm): (**a**) 25%, (**b**) 50%.

**Figure 10 materials-16-03843-f010:**
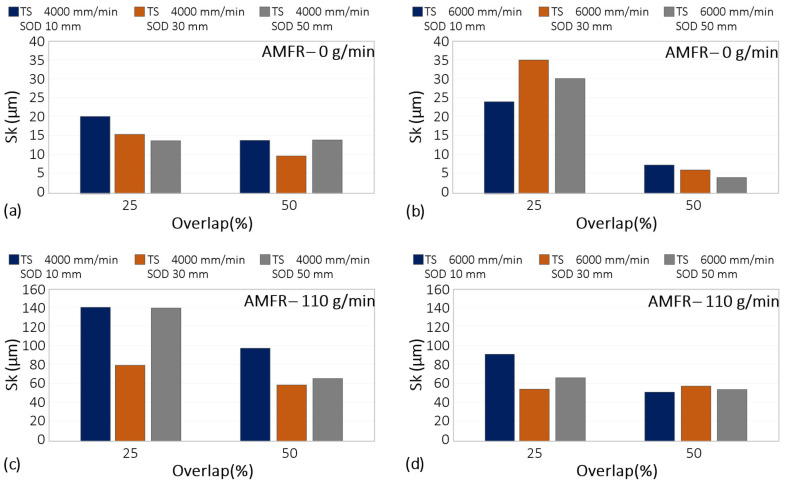
Experimental results for Sk: (**a**) Influence of overlap and SoD for a TS of 4000 mm/min and AMFR 0 g/min; (**b**) Influence of overlap and SoD for a TS of 6000 mm/min and AMFR 0 g/min; (**c**) Influence of overlap and SoD for a TS of 4000 mm/min and AMFR 110 g/min; (**d**) Influence of overlap and SoD for an TS of 6000 mm/min and AMFR 110 g/min.

**Figure 11 materials-16-03843-f011:**
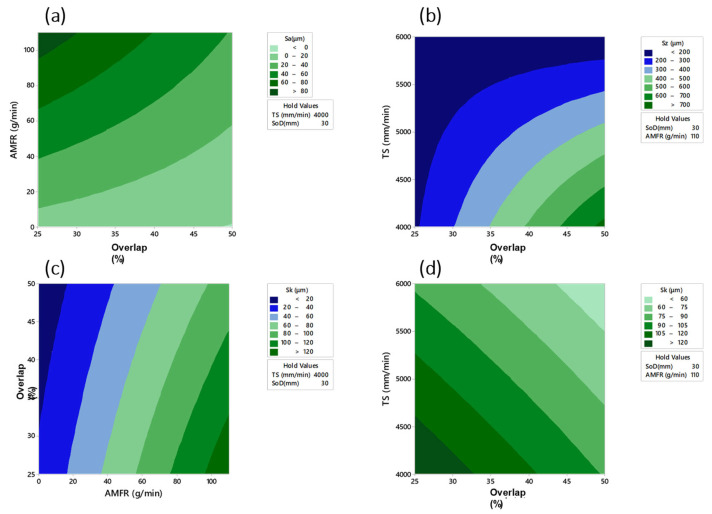
Contour plots based on predictive models relating the main texturing parameters to the results obtained in: (**a**) AMFR vs. Overlap, Sa; (**b**) TS vs. Overlap, Sz; (**c**) AMFR vs. Overlap, Sk; (**d**) Overlap vs. TS, Sk.

**Figure 12 materials-16-03843-f012:**
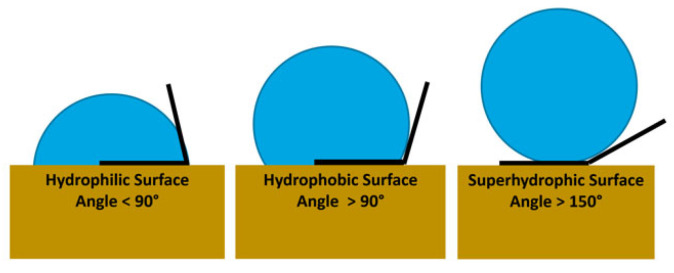
Type of surface obtained as a function of the contact angle of contact when depositing a drop on the modified surface.

**Figure 13 materials-16-03843-f013:**
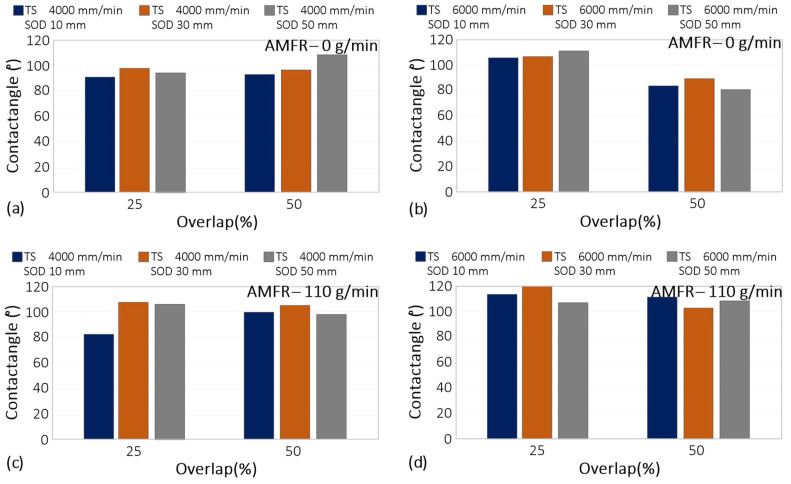
Experimental results for the wettability results: (**a**) Influence of overlap and SoD for a TS of 4000 mm/min and AMFR 0 g/min; (**b**) Influence of overlap and SoD for a TS of 6000 mm/min and AMFR 0 g/min; (**c**) Influence of overlap and SoD for a TS of 4000 mm/min and AMFR 110 g/min; (**d**) Influence of overlap and SoD for a TS of 6000 mm/min and AMFR 110 g/min.

**Figure 14 materials-16-03843-f014:**
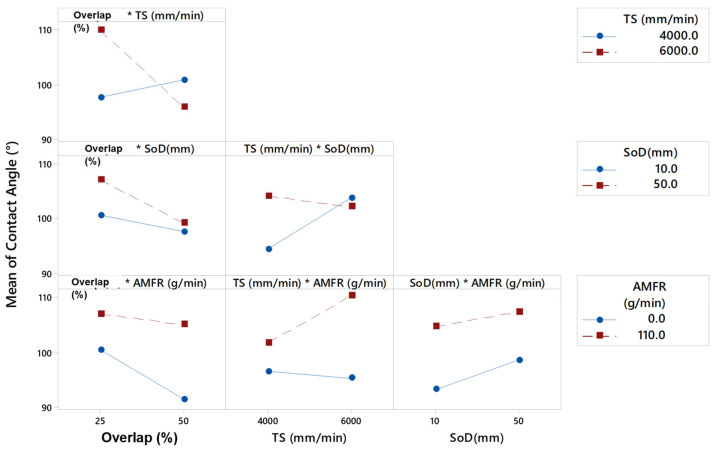
Interaction plots for the parameters TS (mm/min), SoD (mm) and AMFR (g/min).

**Figure 15 materials-16-03843-f015:**
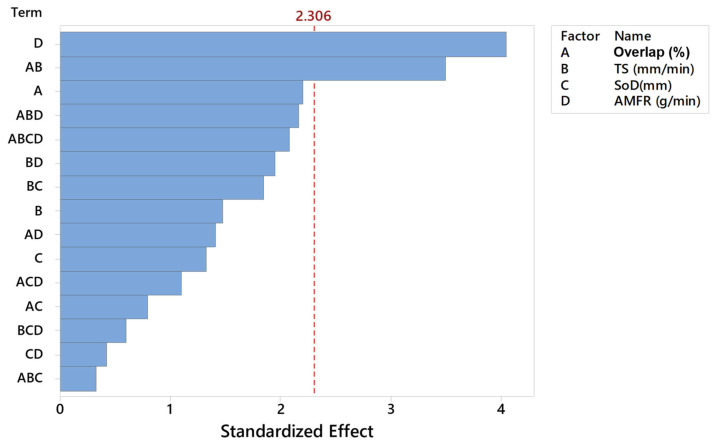
Pareto diagram for different combinations of factors involved in the machining process.

**Figure 16 materials-16-03843-f016:**
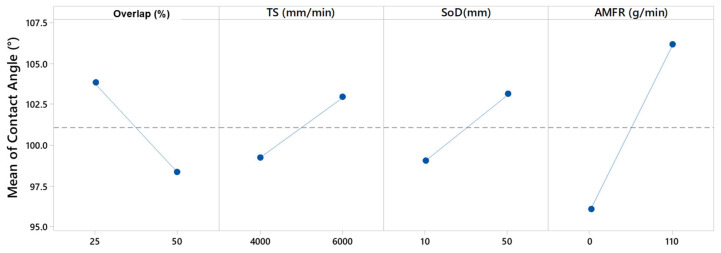
Main effects graph for Overlap (%), TS (mm/min), SoD (mm) and AMFR (g/min).

**Figure 17 materials-16-03843-f017:**
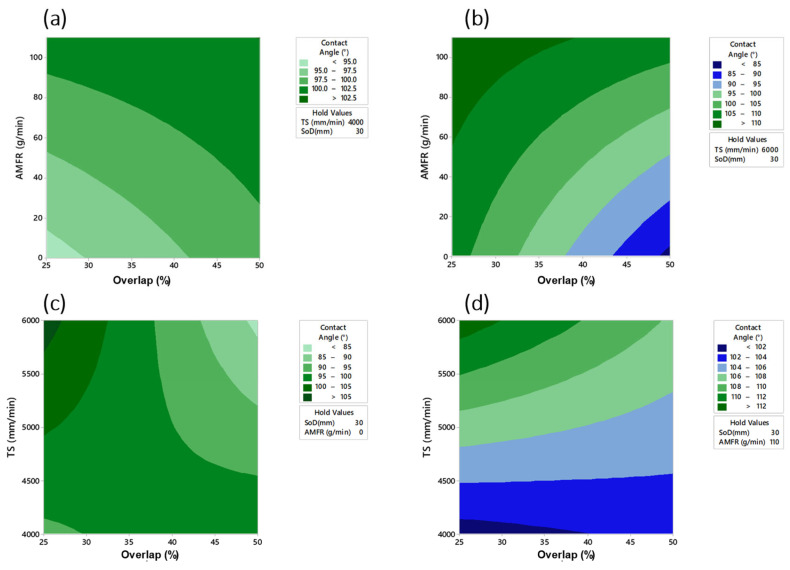
Contour plots for: (**a**) AMFR (g/min) and Overlap (%) for TS = 4000 mm/min and SoD = 30 mm; (**b**) AMFR (g/min) and Overlap (%) for TS = 6000 mm/min and SoD = 30 mm; (**c**) TS (mm/min) and Overlap (%) for SoD = 30 mm and AMFR = 0 g/min; (**d**) TS (mm/min) and Overlap (%) for SoD = 30 mm and AMFR = 110 g/min.

**Table 1 materials-16-03843-t001:** Composition of UNS A92024-T3.

Al	Cu	Mg	Mn	Si	Fe	Zn	Ti	Cr	Others
Rest	3.80–4.90	1.20–1.80	0.30–0.90	≤0.50	≤0.50	≤0.25	≤0.15	≤0.10	≤0.15

**Table 2 materials-16-03843-t002:** Mechanical Properties.

Modulus of Elasticity	73.1 GPa
Hardness, Vickers	137
Ultimate Tensile Strength, UTS	469 MPa
Tensile Yield Strength	324 MPa
Poisson’s Ratio	0.33
Fatigue Strength (R.R Moore Test)	138 MPa

**Table 3 materials-16-03843-t003:** Texturing parameters.

Parameters			
AMFR (g/min)	0		110
Overlap %	25%		50%
SoD (mm)	10	30	50
TS (mm/min)	4000		6000

AMFR, abrasive mass flow rate; SOD, stand-off distance; TS, traverse speed.

**Table 4 materials-16-03843-t004:** Constant parameters.

**Orifice Diameter** **(mm)**	**Focusing Tube Diameter** **(mm)**	**Focusing Tube Length** **(mm)**	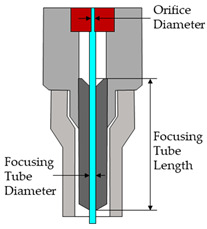
0.25	0.76	380	
**Abrasive Size** **(µm)**	**Abrasive Type**	**Pressure (MPa)**	
500	Garnet	80	

**Table 5 materials-16-03843-t005:** ANOVA analysis showing the statistical influence of texturing parameters on surface quality parameters Sa, Sz and Sk.

	DF	Adj SC	Adj MC	F-Value	*p*-Value
Sa					
Model	10	14,380.6	1438.1	1.87	0.254
Overlap (%)	1	1893.8	1893.8	2.46	0.177
TS (mm/min)	1	1588.1	1588.1	2.07	0.21
SoD(mm)	1	180.3	180.3	0.23	0.649
AMFR (g/min)	1	6186	6186	8.05	0.036
Error	5	3844.5	768.9		
Total	15	18,225.2			
Sz					
Model	10	1,013,053	101,305	8.12	0.016
Overlap (%)	1	238,897	238,897	19.14	0.007
TS (mm/min)	1	329,440	329,440	26.4	0.004
SoD(mm)	1	1670	1670	0.13	0.729
AMFR (g/min)	1	2223	2223	0.18	0.691
Error	5	62,392	12,478		
Total	15	1,075,445			
Sk					
Model	10	29,740.9	2974.1	13.95	0.005
Overlap (%)	1	2861.1	2861.1	13.42	0.015
TS (mm/min)	1	1708	1708	8.01	0.037
SoD(mm)	1	188.1	188.1	0.88	0.391
AMFR (g/min)	1	22,017.1	22,017.1	103.29	0
Error	5	1065.8	213.2		
Total	15	30,806.7			

Colored rows refer to variables that have a statistical significance in the final result.

**Table 6 materials-16-03843-t006:** ANOVA analysis of the texturing parameters on the wettability obtained.

Source	DF	Adj SS	Adj MS	F-Value	*p*-Value
Model	15	2162.33	144.156	3.87	0.03
Overlap (%)	1	180.06	180.062	4.84	0.059
TS(mm/min)	1	80.58	80.582	2.17	0.179
SoD(mm)	1	65.38	65.383	1.76	0.222
AMFR (g/min)	1	610.13	610.132	16.4	0.004
Error	8	297.69	37.211		
Total	23	2460.02			

Colored rows refer to variables that have a statistical significance in the final result.

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
