# Peer review of "Study of Abrasive Water Jet Machining as a Texturing Operation for Thin Aluminium Alloy UNS A92024"

_materials, 2023, doi:10.3390/ma16103843_

Round 1
Reviewer 1 Report
The authors present the use of abrasive water jetting for the surface modification of metal alloys. The authors seem to think that correct combination of high travel speeds at low hydraulic pressures will minimize the power of the water jet and allows the removal of small layers of material. In the results obtained, the influence of the most relevant texturing parameters between hydraulic pressure, traverse speed, abrasive flow and spacing has been determined and allowed a relationship to be established between these variables and surface quality as well as its wettability.
The paper is remarkably interesting, good fits the profile of the Materials journal but the big weakness of this paper is roughness measurements without specifying the measurement conditions and the used filters. In this form, the article is useless.
I also found next errors in this manuscript, and it must be improved.
Noticed errors
1. Lines 55-56. The paragraph is incomplete.
2. Line 120. Why acronym of abrasive flow rate is AMFR, not a AFR?
3. Line 134. What is 50 x 20 x 2 mm3?
4. Lines 150 and 151. On what basis did the authors issue such roughness parameters? It is well known that the Sa parameter poorly characterizes the surface after AWJM.
5. Table 3. Amazing parameters are given here. What is the difference of water nozzle and orifice? They are synonyms. The cutting head consists of a water nozzle (also incorrectly called orifice or jewel) and a focusing tube also called water-abrasive nozzle. The nozzle has a diameter of 250 mm? Please be sure to add a drawing of it!!!
6. Chapter 2.1. No information about mechanical properties of machining material
7. Chapter 2.1. No information about the abrasive material. What does the grain size of 500 um mean? What is the grain distribution? What is the shape of the grains? What kind of garnet was used: alluvial or crushed rock? From what deposit: country/region? What are the basic properties of the garnet used? The characteristic of the nozzle is not its diameter, but the inner diameter (ID)
8. Figure 14 rather not is Pareto diagram, but figure 13 looks more like a Pareto chart.
9. The big weakness of this paper is roughness measurements without specifying the measurement conditions and the used filters. In this form, the article is useless.
10. Adding conclusions for further research would certainly increase the value of the paper.
Author Response
Please read the responses to the revisions in the attached .pdf file.

Reviewer 2 Report
The article is aimed at the current topic of processing specific surfaces with (abrasive) water jet technology, which is interesting both from the point of view of research and practice. The authors performed a series of planned experiments and evaluated them using statistical methods. They also created models allowing to determine the roughness and wettability parameters of the machined surface at different values of the process parameters.
The presentation of the results and their interpretation contains several shortcomings, the elimination of which is crucial for assessing the correctness of the interpretation of the measured results.
1. Fig. 3 - it is not clear which of the graphs is a), which is b), c), d), e), as indicated in the description of the figure. There are only 4 graphs, not 5. Even according to the legends of the individual graphs, it is not clear what the difference was between the parameters when creating the samples evaluated in the graphs located on the left part of the figure since the legends are identical. The same holds for the graphs on the right side of the figure. Thus, the entire image is incomprehensible, and this problem also applies to the graphs and their descriptions in Figures 6; 8 and 11.
2. To Fig. 4 the scale is missing, the description in the left part of Fig. is unreadable. This problem also concerns Fig. 7.
3. In line 318, there is a typo in the index of the parameter Rz, which is erroneously listed there as Rt.
4. In Fig. 9 a) the values of Sa, which have negative values, are shown in light green. The authors must explain their predictive model, as such values are in principle not achievable and indicate the incorrectness of the created model.
5. In Fig. 12, the mean values of the measured contact angles are presented. Still, the description of the statistical file is missing - from how many measured values the value was determined and with what standard deviation, for example.
6. What is the difference between the terms simultaneously used in the article: travel speed and traverse speed?
Based on the above findings, I do not recommend the article in its current form for publication.
Author Response

(The authors gave the same response as above.)

Reviewer 3 Report
The current research article titled: Study of abrasive water jet machining as a texturing operation for thin aluminium alloy UNS A92024 pertains to an interesting topic with practical value as well. Although conceptually is very interesting, I am afraid that it has a number of significant flaws, especially in the discussion part and regarding the deduced conclusions.
More specifically:
· -a minor improvement in syntax and grammar is suggested, while a number of typos have to be corrected.
· -authors have to provide some more details regarding the measuring methods. For example, in measuring the Surface Roughness and the Surface Wettability were any ISO standards followed? How many measurements were taken? Please present in more details the measuring methods.
· -authors state: "In general terms, at a TS of 6000 mm/min, the greater the distance to the workpiece and the higher the percentage of overlap, the better the surface quality in terms of Sa.". Nevertheless, based on the respective diagram, the Sa values for TS 6000 mm/min, SOD 30 and 50 mm and Overlap 25% are higher than the respective for SOD 10, hence, the aforementioned conclusion is not correct. Please revise accordingly and be very careful in any deduced conclusion in order to avoid any misconceptions and false conclusions.
· -again, in lines 219 - 227 authors reach conclusions that are not fully supported by the respective data. Please revise and become more clear. Moreover, authors have to discuss and explain why for TS 4000 mm/min, SOD 50 mm and Overlap 25% when abrasive is utilized there is such a significant increase in Sa.
· -authors state: "It is noteworthy that similar Sz values are obtained in both abrasive and non-abrasive conditions.". Again, this statement it is not supported by the data of Figure 6. During the whole discussion and results' presentation authors come up with some very questionable conclusions. Thus, in my humble opinion, the whole discussion section has to be carefully revised.
· -authors state: "The fit of the models is 78.91% for the Sa parameter, 94.20% for the Sz parameter and 96.54% for the Sk parameter.". Models have to be presented in details.
· -authors state: "In general, it appears that a 25% overlap favors the formation of smaller contact angles when interacting with the three input variables.", a statement that it is entirely opposite to the results of Figure 12.
Considering the aforementioned, I am afraid that the current paper cannot be accepted in its present form.
Moderate editing of English language is suggested
Author Response

(The authors gave the same response as above.)

Round 2
Reviewer 1 Report
The authors corrected the noticed errors and in this form the article is suitable for printing.
Author Response
We thank the reviewer for his comments to improve the final quality of the article.
Reviewer 2 Report
The authors fixed mentioned inconsistencies and mistakes and also explained the missing details of their research. In my opinion, the article is a good contribution to the topic.
English is not my native language. In my opinion, the language level after unifying several labels for one term is good, but it still needs a final check.Author Response
We thank the reviewer for his comments to improve the final quality of the article.
Reviewer 3 Report
Authors have significantly improved the manuscript according to the reviewers' comments and suggestions. Considering the aforementioned improvement, in my opinion, the current article can be accepted in its current form, and only some minor issues have to be addressed. Namely, it is suggested to add error bars in all the diagrams and it would be useful if authors explain why in equations 1-3 have included terms that based on Table 5 are not significant.
Author Response
We thank the reviewer for his comments and advice. The following answers to the queries are given below:
The methodology for evaluating the surface quality in terms of Sa, Sz and Sk has been carried out based on the evaluation of a constant surface area obtained after texturing. Repeated measurements of the same area generate the same results. The evaluation methodology under the standard established in the methodology section allows us to obtain a value of these parameters for the evaluation area. The equipment used for this evaluation is correctly calibrated and can carry out these evaluations under the indicated standard. For this reason, no error bars have been shown.
About the evaluation of the contact angle, the deviation obtained has been indicated in the text. This value is also very small and including it in the graph may make it difficult to understand and hardly visible.
The values that are not significant have been kept in the predictive models because, although their variation is not statistically significant, their application improves the fit of the model, albeit by a very small percentage. The authors consider that it is of interest to present the best fit for future applications and continuations of this work.